



# The Lifetime of Nitrogen Oxides in an Isoprene Dominated Forest

Paul S. Romer[1], Kaitlin C. Duffey[1], Paul J. Wooldridge[1], Hannah M. Allen[2,3], Benjamin R. Ayres[2], Steven S. Brown[4], William H. Brune[5], John D. Crounse[6], Joost de Gouw[4,7], Danielle C. Draper[2,8], Philip A. Feiner[5], Juliane L. Fry[2], Allen H. Goldstein[9,10], Abigail Koss[4,7], Pawel K. Misztal[10], Tran B. Nguyen[6,11], Kevin Olson[9], Alex P. Teng[6], Paul O. Wennberg[6,12], Robert J. Wild[4,7], Li Zhang[5], and Ronald C. Cohen[1,13]

[1]Department of Chemistry, University of California at Berkeley, Berkeley, CA, USA
[2]Department of Chemistry, Reed College, Portland, OR, USA
[3]Division of Chemistry and Chemical Engineering, California Institute of Technology, Pasadena, CA, USA
[4]Chemical Sciences Division, Earth System Research Laboratory, National Oceanic and Atmospheric Administration, Boulder, CO, USA
[5]Department of Meteorology, Pennsylvania State University, University Park, PA, USA.
[6]Division of Geological and Planetary Sciences, California Institute of Technology, Pasadena, CA, USA
[7]Cooperative Institute for Research in Environmental Sciences, University of Colorado, Boulder, CO, USA
[8]Department of Chemistry, University of California, Irvine, CA, USA
[9]Department of Civil and Environmental Engineering, University of California at Berkeley, Berkeley, CA, USA
[10]Department of Environmental Science, Policy and Management, University of California at Berkeley, Berkeley, CA, USA
[11]Department of Environmental Toxicology, University of California, Davis, CA, USA
[12]Division of Engineering and Applied Science, California Institute of Technology, Pasadena, CA, USA
[13]Department of Earth and Planetary Sciences, University of California at Berkeley, Berkeley, CA, USA

*Correspondence to:* Ronald C. Cohen (rccohen@berkeley.edu)





**Abstract.** The lifetime of $NO_x$ ($NO_x \equiv NO + NO_2$) affects the concentration and distribution of $NO_x$ and the spatial patterns of nitrogen deposition. Despite its importance, the lifetime of $NO_x$ is poorly constrained in rural and remote continental regions. We use measurements from a site in central Alabama during the Southern Oxidant and Aerosol Study (SOAS) in summer 2013 to provide new insights into the chemistry of $NO_x$ and $NO_x$ reservoirs. We find that the lifetime of $NO_x$ during the daytime is

controlled primarily by the production and loss of alkyl and multifunctional nitrates ($\Sigma ANs$). During SOAS, $\Sigma AN$ production was rapid, averaging 90 ppt hr$^{-1}$ during the day, and occurred predominantly during isoprene oxidation. Analysis of the $\Sigma AN$ and $HNO_3$ budgets indicate that $\Sigma ANs$ have an average lifetime of under 2 hours, and that approximately 45% of the $\Sigma ANs$ produced at this site are rapidly hydrolyzed to produce nitric acid. We find that $\Sigma AN$ hydrolysis is the largest source of $HNO_3$ and the primary pathway to permanent removal of $NO_x$ from the boundary layer in this location. Using these new constraints

on the fate of $\Sigma ANs$, we find that the $NO_x$ lifetime is $11 \pm 5$ hours under typical midday conditions. The lifetime is extended by storage of $NO_x$ in temporary reservoirs, including acyl peroxy nitrates and $\Sigma ANs$.

## 1   Introduction

The concentration and chemistry of nitrogen oxides ($NO_x$) in Earth's troposphere has a significant and non-linear effect on the oxidative capacity of the atmosphere. This in turn affects the production, composition, and aging of aerosols and the lifetime

of greenhouse gases such as methane. Concentrations of $NO_x$ control the production of ozone, a respiratory health hazard, important oxidant, and greenhouse gas. In addition, the deposition of reactive nitrogen is an important source of nutrients in some ecosystems (e.g. Fowler et al., 2013).

$NO_x$ is emitted by both anthropogenic and biogenic sources, including motor vehicles, power plants, forest fires, and soil bacteria (e.g. Dallmann and Harley, 2010; Mebust and Cohen, 2014; Hudman et al., 2012), and is temporarily or permanently

removed from the atmosphere by chemical conversion to higher oxides of nitrogen, e.g., $R(O)OONO_2$, $RONO_2$, and $HNO_3$. Across much of the globe, the balance of these sources and sinks is in a period of dramatic change, with large reductions of $NO_x$ emissions occurring in North America and Europe and significant increases occurring in Asia (e.g. Russell et al., 2012; Curier et al., 2014; Reuter et al., 2014). Understanding the effects of changes in $NO_x$ emissions on the concentration and spatial distribution of $NO_x$ requires detailed knowledge of the chemistry and transport of $NO_x$ and $NO_x$ reservoirs. These

reservoirs are poorly understood and represent a significant uncertainty in analyses of $NO_x$ emissions and ozone production (e.g. Ito et al., 2007; Browne and Cohen, 2012; Mao et al., 2013).

The net chemical loss of $NO_x$ is difficult to directly observe. Observational methods for determining the lifetime of $NO_x$ are easiest to apply in the outflow of isolated emissions, where the declining concentration of $NO_x$ or the changing ratio of $NO_x$ to total reactive nitrogen ($NO_y$) provide clear evidence for $NO_x$ loss (e.g. Ryerson et al., 1998; Dillon et al., 2003; Alvarado

et al., 2010; Valin et al., 2013). In rural and remote regions, emissions and concentrations of $NO_x$ and $NO_y$ are typically slowly varying over large distances (e.g. Browne et al., 2013), preventing the loss of $NO_x$ from being directly observable. Nor can the lifetimes found in plume studies be easily translated into an appropriate lifetime in the regional background. Short-lived $NO_x$ reservoirs such as Peroxy Acyl Nitrate (PAN) can efficiently remove $NO_x$ in a plume, but act as a source of $NO_x$ in rural





and remote regions (Finlayson-Pitts and Pitts, 1999). In addition, the non-linear interactions between $NO_x$ and OH make the lifetime of $NO_x$ in a fresh plume very different from its lifetime several hours downwind (e.g. Martinez et al., 2003; Valin et al., 2013).

To constrain the lifetime of $NO_x$ in rural and remote regions, observations of reactive nitrogen species must be combined with
an understanding of the chemical transformations between $NO_x$ and its higher oxides. If the production, loss, and fate of these higher oxides are accurately understood, then the lifetime of $NO_x$ can be calculated by tracing the flow of reactive nitrogen through the system. Here, we evaluate the daytime lifetime of $NO_x$ in the rural southeast United States, using measurements taken from 1 June – 15 July 2013 as part of the Southern Oxidant and Aerosol Study (SOAS). In-situ measurements of Volatile Organic Compounds (VOCs), atmospheric oxidants, and a wide range of reactive nitrogen compounds are used to determine
the production and loss rates for nitric acid, alkyl and multifunctional nitrates, and peroxy nitrates. These rates are used to assess the lifetime of $NO_x$ in this region.

## 2   The $NO_y$ family and the lifetime of $NO_x$

During the day, $NO_x$ is lost by associating with other radicals to produce higher oxides of nitrogen, primarily nitric acid ($HNO_3$), alkyl and multifunctional nitrates ($\Sigma ANs = \Sigma RONO_2$), and peroxy nitrates ($\Sigma PNs = \Sigma R(O)OONO_2$) (e.g. Day
et al., 2003; Perring et al., 2010). The sum of these and other higher oxides such as $N_2O_5$ and HONO are collectively known as $NO_z$ ($NO_z \equiv NO_y - NO_x$).

$NO_x$ is oxidized to produce the major daytime classes of $NO_z$ through reactions (R1), (R2b), and (R3).

$$NO_2 + OH + M \rightarrow HNO_3 + M \tag{R1}$$

$$NO + RO_2 + M \rightarrow RONO_2 + M \tag{R2b}$$

$$NO_2 + R(O)O_2 + M \rightarrow R(O)OONO_2 + M \tag{R3}$$

$NO_x$ can also be converted to $NO_z$ through reactions of the $NO_3$ radical. Although these reactions are most important at night, previous studies have shown that $NO_3$ chemistry can produce $NO_z$ during the day if concentrations of alkenes are high (e.g.
Fuentes et al., 2007; Mogensen et al., 2015; Ayres et al., 2015).

The production and fate of different $NO_z$ species determine the lifetime of $NO_x$. Some of these species are short-lived and re-release $NO_x$ back to the atmosphere within hours of being formed. If the lifetime for the conversion of a $NO_z$ species back to $NO_x$ is shorter than typical $NO_x$ lifetimes in the atmosphere, then $NO_x$ and these $NO_z$ species interact, and their concentrations will approach a steady-state ratio. As $NO_x$ is removed from the system, some of the short-lived $NO_z$ species
dissociate, buffering the concentration of $NO_x$. In this way, the presence of $NO_x$ reservoirs directly extends the lifetime of $NO_x$.



One method to take this buffering into account when calculating the lifetime of $NO_x$ is to consider the sum of $NO_x$ and all $NO_z$ species with lifetimes to re-release of $NO_x$ shorter than the atmospheric lifetime of $NO_x$. We define this sum as short-lived reactive nitrogen, or $NO_{SL}$. The remaining forms of reactive nitrogen are defined as long-lived reactive nitrogen ($NO_{LL}$). We provisionally use a cutoff of 7 hours to divide $NO_z$ species between $NO_{SL}$ and $NO_{LL}$, based on estimated $NO_x$ lifetimes

determined from satellite observations of $NO_2$ (e.g. Lamsal et al., 2010; Valin et al., 2013). In areas well removed from large $NO_x$ sources, $NO_x$ and its short-lived reservoirs interconvert significantly faster than the rate of change of $NO_x$. Under these conditions, the lifetime of $NO_x$ ($\tau_{NO_x}$) is equal to the lifetime of $NO_{SL}$. If the conversion of $NO_{LL}$ to $NO_{SL}$ is negligible, then the lifetime of $NO_x$ can be calculated by Eq. (1):

$$\tau_{NO_x} = \tau_{NO_{SL}} = \frac{[NO_{SL}]}{\mathcal{L}(NO_{SL})} \tag{1}$$

Throughout this paper, we use $\mathcal{L}(X)$ to indicate the gross loss rate of the compound or class of compounds X.

The relationship and interactions between $NO_{SL}$ and $NO_{LL}$, and their typical compositions in the planetary boundary layer, are shown in Fig. 1. In the summertime at midlatitudes, peroxy nitrates typically release $NO_x$ within hours of being formed (LaFranchi et al., 2009), making them a component of $NO_{SL}$. Under these same conditions, nitric acid typically converts back to $NO_x$ on timescales of 100 hours or greater (Finlayson-Pitts and Pitts, 1999) and is a component of $NO_{LL}$. The fate and

lifetime of $\Sigma ANs$, the third major component of $NO_z$, remain poorly understood, making it uncertain whether $\Sigma ANs$ act as a component of $NO_{SL}$ or $NO_{LL}$ (Perring et al., 2013, and references therein). This is especially true for the multifunctional, biogenically-derived nitrates that are the primary component of $\Sigma ANs$ in forested areas (e.g. Beaver et al., 2012).

Recent studies of multifunctional nitrates suggest that the main daytime loss pathways of these species are deposition, reaction with OH, photolysis, and heterogeneous hydrolysis to produce nitric acid (e.g. Darer et al., 2011; Browne et al., 2013;

Lee et al., 2014; Müller et al., 2014; Lee et al., 2015; Nguyen et al., 2015). These recent studies, combined with the extensive measurements made during SOAS, allow us to provide new constraints on the lifetime and fate of $\Sigma ANs$ and therefore to more accurately determine the lifetime of $NO_x$.

## 3   Instrumentation and measurements

The primary ground site for SOAS was located in Bibb County, Alabama ($32.90289°$ N, $87.24968°$ W) at the Centreville (CTR)

long-term monitoring site in the SouthEastern Aerosol Research and CHaracterization (SEARCH) Network (Hansen et al., 2003). This location is 40 km southeast of Tuscaloosa (population 95,000), and 90 km southwest of Birmingham (population 210,000). Comparison with long-term measurements indicate that the summer of 2013 was cooler and cloudier than typical for previous summers (Hidy et al., 2014). Gas-phase measurements used in this study were located on a 20 m walk-up tower at the edge of the forest. Nitrate ion and meteorological parameters were measured in a clearing approximately 50 m away from

the tower.

A nearly complete suite of reactive nitrogen species, including NO, $NO_2$, $\Sigma PNs$, $\Sigma ANs$, $HNO_3$, and $NO_3^-$, was measured during SOAS. NO was measured using the chemiluminescence instrument described in Min et al. (2014). The reaction of





ambient NO with added excess $O_3$ formed excited $NO_2^*$ molecules. A fraction of these fluoresce, and the emitted photons were collected on a red-sensitive photomultiplier tube (Hamamatsu H7421-50). Calibrations were performed every 2 hours by diluting NO standard gas (5.08 ppm $\pm5\%$ NO in $N_2$, Praxair) to 3–20 ppb in zero air and adding it to the instrument inlet. The mixing ratio was corrected for enhanced quenching by water vapor (Thornton et al., 2000) using co-located measurements of

relative humidity and temperature.

$NO_2$, $\Sigma$PNs, and $\Sigma$ANs were measured via Thermal Dissociation Laser Induced Fluorescence (TD-LIF), as described by Day et al. (2002). Ambient air was drawn into a multipass White cell, where a 532 nm Nd-YAG laser excited the $NO_2$ molecules, and their fluorescence signal was collected on a photomultiplier tube (Hamamatsu H7421-50). The same instrument was used to measure the sum of peroxy nitrates and the sum of alkyl and multifunctional nitrates by first passing the air through

a heated oven, where the organic nitrates dissociated to form $NO_2$. A fourth channel in the TD-LIF instrument measured the concentration of alkyl and multifunctional nitrates in the aerosol phase (Rollins et al., 2010). All 4 channels were calibrated by injecting $NO_2$ standard gas (5.03 ppm $\pm5\%$ $NO_2$ in $N_2$, Praxair) and corrected for enhanced quenching by water vapor.

Nitric acid was measured in the gas phase by chemical ionization mass spectrometry, using $CF_3O^-$ as the reagent ion (Crounse et al., 2006). The ions were quantified using a compact time-of-flight mass spectrometer, and the instrument was

calibrated in the field using isotopically labeled nitric acid. Particle-phase inorganic nitrate ($NO_3^-$) was measured using a Monitor for AeRosols and GAses (MARGA) (Allen et al., 2015). Ambient air was drawn through a rotating wet-walled denuder which collected water soluble gas-phase compounds. Particle-phase compounds were captured by a steam-jet aerosol collector downstream of the denuder. Water soluble ions from both phases were then quantified via ion chromatography.

Measurements of reactive nitrogen species are summarized in Fig. 2. Concentrations of $NO_{SL}$ compounds (NO, $NO_2$, and

$\Sigma$PNs) are shown in Fig. 2a. Afternoon concentrations of $NO_2$ and NO were typically around 220 ppt and 50 ppt respectively. After sunset, NO dropped to near zero, and $NO_2$ began to increase. At sunrise, NO concentrations rapidly rose to over 200 ppt between 6 am and 8 am Central Standard Time (CST) while $NO_2$ decreased sharply. By 11 am, when the daytime boundary layer was well developed, the concentrations of NO and $NO_2$ returned to their typical afternoon values. Concentrations of $\Sigma$PNs were 160 ppt at sunrise, increased to a maximum concentration of 300 ppt at 9 am and declined slowly throughout the

rest of the day.

Concentrations of $NO_3^-$ and $HNO_3$, components of $NO_{LL}$, are shown in Fig. 2b. Both species increased slowly after sunrise and reached a maximum combined concentration of 300 ppt at 1 pm, before declining to a combined concentration of 175 ppt at night. Concentrations of $\Sigma$ANs, whose partitioning into $NO_{SL}$ and $NO_{LL}$ is not known, are shown in Fig. 2c. $\Sigma$ANs averaged 150 ppt during the night and increased sharply after sunrise. After reaching a maximum of 250 ppt at 8 am, $\Sigma$ANs

declined slowly to a minimum concentration of 125 ppt at sunset.

$HO_x$ radicals ($HO_x \equiv OH + HO_2$) and OH Reactivity were measured via Fluorescence Assay by Gas Expansion (FAGE) of OH. A 308 nm dye laser excited the OH radicals and their fluorescence was detected by an electronically gated microchannel plate detector (Faloona et al., 2004). Calibration of the system was performed by in-situ generation of OH radicals via photolysis of water vapor. Chemical zeroing was performed by periodically adding $C_3F_6$ to the sampling inlet in order to quantify

the interference from internally generated OH observed in previous field campaigns (Mao et al., 2012). $HO_2$ was measured in



a second channel by adding NO to chemically convert $HO_2$ to OH. The amount of added NO was regulated such that $HO_2$ but not $RO_2$ was converted to OH (Fuchs et al., 2011). Total OH Reactivity (OHR) was measured by drawing ambient air through a flow tube and mixing it with a fixed concentration of OH. At the end of the flow tube, the concentration of OH was measured. The OH Reactivity is determined by the slope of the OH signal versus reaction time (Mao et al., 2009).

Measured concentrations of OH peaked at 0.045 ppt and concentrations of $HO_2$ at 30 ppt during SOAS (Fig. 3). Both OH and $HO_2$ increased slowly throughout the morning and reached their maximum in the early afternoon. Concentrations then fell as the sun set, with OH usually dropping below 0.01 ppt by 7 pm. The measured OH Reactivity was high, reaching an afternoon peak of close to 25 $s^{-1}$ (Fig. 3). OHR decreased throughout the night, reaching a minimum of 10 $s^{-1}$ just before sunrise.

Volatile organic compounds were measured primarily by Gas Chromatography - Mass Spectrometry (GC-MS). Samples were collected in a liquid-nitrogen cooled trap for five minutes, and then transferred by heating onto an analytical column, and detected using an electron-impact quadrupole mass-spectrometer (Goldan et al., 2004; Gilman et al., 2010). This system was able to quantify a wide range of compounds including alkanes, alkenes, aromatics, isoprene, and multiple monoterpenes. The sum of methyl vinyl ketone (MVK) and methacrolein (MACR) was measured using a Proton Transfer Reaction - Time of

Flight - Mass Spectrometer (PTR-TOF-MS) (Kaser et al., 2013). The interference in this measurement from the decomposition of isoprene hydroperoxides on instrument inlets (Rivera-Rios et al., 2014) is not corrected for, and increases the uncertainty in this measurement by approximately 20%.

    VOC measurements at the site show that the OHR was dominated by reaction with biogenic compounds. Figure 3 shows the OH Reactivity of individually measured compounds as a stacked area plot. In the daytime, isoprene accounted for nearly half

of the total reactivity, while VOCs typically attributed to anthropogenic activities, including alkanes, aromatics, and simple alkenes, were responsible for less than 10% of the measured OHR. Not included in Fig. 3 is the reactivity of VOCs whose reaction with OH does not lead to net loss of OH, and therefore does not contribute to the measured OHR. These compounds, primarily isoprene hydroperoxides and C5-hydroxyaldehydes, have an average daytime reactivity of 2 $s^{-1}$. The sum of individual reactivities shows a similar diurnal pattern to the measured OHR, and accounts for 70–85% of the total. Unknown

biogenic emissions, small aldehydes and alcohols, and other 2nd and 3rd generation VOC oxidation products are all possible contributors to the missing reactivity (e.g. Di Carlo et al., 2004; Goldstein and Galbally, 2007; Pusede et al., 2014).

    Measurements of ozone were made using a Cavity Ring Down Spectrometer (Washenfelder et al., 2011). $O_3$ is chemically converted to $NO_2$ by reaction with excess NO, and the resulting $NO_2$ is measured by cavity ring-down spectroscopy at 404 nm. Meteorological parameters including temperature and solar radiation were measured by Atmospheric Research and Analysis

as part of SEARCH.





## 4 The production and loss of individual NO$_x$ reservoirs

### 4.1 Nitric acid

In the boundary layer, the production of nitric acid is typically followed by deposition and thus leads to the permanent removal of reactive nitrogen from the atmosphere. Nitric acid can undergo photolysis or reaction with OH to produce NO$_x$, but these
processes are slow (Burkholder et al., 1993; Atkinson et al., 2006), with an average calculated rate during SOAS of less than 0.2 ppt hr$^{-1}$. Gas-phase nitric acid can also partition into aerosols. Nitric acid is long lived in the particle phase and is typically lost by re-evaporation into the gas phase (e.g. Hennigan et al., 2008). The loss of nitric acid through deposition of aerosols is typically negligible compared to its gas-phase deposition (e.g. Zalakeviciute et al., 2012). Because HNO$_3$ releases NO$_x$ so slowly, it is a component of NO$_{LL}$.

The deposition velocity (v$_{dep}$) of HNO$_3$ in the gas phase was measured during SOAS by Nguyen et al. (2015). Around midday, when the boundary layer is well developed, the deposition velocity can be combined with the boundary layer height (BLH) to calculate a loss rate of HNO$_3$:

$$\mathcal{L}(\text{HNO}_3) = \frac{\text{BLH}}{\text{v}_{dep}} \cdot [\text{HNO}_3] \tag{2}$$

Using this method, we find the lifetime of HNO$_3$(g+p) to be 6 hours at noon. In the late afternoon, changing boundary layer
dynamics make this calculation of the loss rate inaccurate (e.g. Papale et al., 2006; Millet et al., 2015). The loss of nitric acid in the late afternoon was therefore calculated by fitting periods of consistent decay between 3 pm and 7 pm with an exponential curve. By fitting only the periods of consistent decay, we aim to select for periods where the production of nitric acid is at a minimum and the observed net decay of HNO$_3$ is similar to its gross loss rate. Because nitric acid reversibly partitions between the gas and particle phases, the lifetime was calculated based on the concentration of nitric acid in both phases. The lifetime of
HNO$_3$(g+p) using the afternoon decay is $5^{+3}_{-2}$ hours, similar to the lifetime of HNO$_3$(g+p) calculated using Eq. 2 at noon.

Using this loss rate of HNO$_3$, the production rate necessary to sustain the concentrations of nitric acid observed during SOAS can be calculated (Fig. 4). This inferred production rate for each hour is defined as the difference between the rate of change in the concentration of HNO$_3$ and the loss rate. The rate of change of HNO$_3$ was determined as the slope of a best-fit line of the concentration of HNO$_3$ versus time for each hour. Due to the variability in wind direction during SOAS and the lack
of large NO$_x$ sources near the CTR site, we assume that the inferred source of nitric acid is in-situ chemical production and not distant production followed by long range transport. The inferred source also shows little variation with wind direction, further evidence that the inferred source represents local production of nitric acid. The changing boundary layer height could significantly impact the inferred production rate of HNO$_3$ during the early morning, but it is likely unimportant at midday. Also shown in Fig. 4 is the rate of nitric acid production from the reaction of OH+NO$_2$ (R1), using the rate constant measured
by Mollner et al. (2010). The vertical bars for the inferred rate represent the combined effects of the uncertainty in both the fit of [HNO$_3$] v. time and in the calculated HNO$_3$ lifetime, as well as the day-to-day variations in the observations. The vertical bars shown for the production of nitric acid from the OH+NO$_2$ reaction include both the systematic and random errors in the measurements of OH and NO$_2$ and in the rate coefficient, $k_{\text{OH}+\text{NO}_2}$, combined in quadrature.



Between 10 am and 2 pm, when photochemistry is most active, the inferred production rate is 3–4 times larger than the rate of $OH + NO_2$ (R1), a difference of approximately 30 ppt hr$^{-1}$. The most likely explanation for the missing $HNO_3$ production during this time is the heterogeneous hydrolysis of $\Sigma ANs$. This has been proposed as an important source of nitric acid over the Canadian boreal forest (Browne et al., 2013), and the hydrolysis of tertiary alkyl nitrates on atmospherically relevant timescales has been observed in several laboratory experiments (e.g. Darer et al., 2011; Liu et al., 2012; Rindelaub et al., 2015). If $\Sigma ANs$ are being converted to nitric acid, this process should appear as a sink in the budget of $\Sigma ANs$.

## 4.2 Alkyl and multifunctional nitrates

Previous observational studies have found that the production of $\Sigma ANs$ is rapid in forested regions (e.g. Day et al., 2009; Beaver et al., 2012; Fry et al., 2013; Browne et al., 2013), but the subsequent fates of these biogenic nitrates are not well constrained. During the day, $\Sigma ANs$ are produced primarily from the reaction of organic peroxy radicals ($RO_2$) with NO. Most of the time, this leads to the formation of RO and $NO_2$ (R2a), but a fraction of the time produces an organic nitrate (R2b). The branching ratio $k_{R2b}/(k_{R2b} + k_{R2a})$ is designated $\alpha$ and varies with the structure of the R group, as well as the temperature and pressure.

$$NO + RO_2 \rightarrow RO + NO_2 \tag{R2a}$$

$$NO + RO_2 + M \rightarrow RONO_2 + M \tag{R2b}$$

Organic peroxy radicals are produced in the daytime troposphere predominantly by the reaction of OH with VOCs and are lost through reaction with NO, $HO_2$, and $RO_2$, or through unimolecular isomerization. These radicals reach steady state within seconds, allowing the production of $\Sigma ANs$ via reaction (R2b) to be calculated as:

$$P(\Sigma ANs) = \sum_{R_i} \alpha_i \cdot f_{NO_i} \cdot k_{OH+R_i} \cdot [R_i] \cdot [OH] \tag{3}$$

The value $f_{NO}$ represents the fraction of $RO_2$ radicals that are lost by reaction with NO. This value was calculated separately for each measured VOC and is equal to the rate of reactions R2b and R2a, divided by the sum of all $RO_2$ loss rates. Rate constants for the reaction of $RO_2$ radicals with NO, $HO_2$, and other $RO_2$ radicals are taken from the Master Chemical Mechanism v3.2 (Saunders et al., 2003) for all species other than isoprene and methacrolein. The reactions of isoprene-derived $RO_2$ radicals are based on the LIM-1 scheme described by Peeters et al. (2014), with the rate of unimolecular isomerization scaled to match the rate of HPALD formation observed in chamber experiments by Crounse et al. (2011). For methacrolein, we include the isomerization rate described by Crounse et al. (2012). Unimolecular isomerization is not included for any other $RO_2$ species. Concentrations of $RO_2$ radicals are calculated iteratively at each point until they converge.

Values of $k_{OH+R_i}$ and $\alpha_i$ are taken from Atkinson and Arey (2003) and Perring et al. (2013) respectively, with the following exceptions. An $\alpha$ of 0.26 is used for $\alpha$-pinene, following Rindelaub et al. (2015). An $\alpha$ of 0.12 is used for isoprene. This is in





the middle of the range of branching ratios for isoprene (0.09–0.15) found in recent experiments (e.g. Paulot et al., 2009; Teng et al., 2015; Xiong et al., 2015).

The missing OH Reactivity (Fig. 3) is included in this calculation as a generic VOC that forms $RO_2$ radicals that react with the same kinetics as $CH_3CH_2O_2$. Box model calculations using chemistry from the MCMv3.2 and a modified version of the UW-CAFE model (Saunders et al., 2003; Wolfe and Thornton, 2011) suggest that the missing OHR can be explained by the reactions of unmeasured second and later generation oxidation products. While values of $\alpha$ for these compounds are not known, measured branching ratios for other highly oxidized compounds are typically less than 0.01. We therefore assume that the missing OHR has an effective $\alpha$ value of 0.005.

The daytime production of $\Sigma$ANs also includes a minor contribution from the reaction of $NO_3$ with alkenes, via reactions (R4) and (R5).

$$NO_2 + O_3 \rightarrow NO_3 + O_2 \tag{R4}$$

$$NO_3 + R \rightarrow RONO_2 \tag{R5}$$

Concentrations of isoprene and monoterpenes were sufficiently elevated during SOAS that reaction with these compounds is a significant fraction of the total daytime loss of $NO_3$. Calculations following Ayres et al. (2015) indicate that this pathway produces $\Sigma$ANs at an average rate of 10 ppt hr$^{-1}$.

The calculated total rate of $\Sigma$AN production via (R2b) and (R5) is rapid, averaging approximately 90 ppt hr$^{-1}$ between 8 am and 4 pm (Fig. 5). The oxidation of isoprene accounts for over three-quarters of the production of $\Sigma$ANs, and monoterpenes account for an additional 15%. Based on the uncertainty in each term in Eq. 3, the total systematic uncertainty in the production rate of $\Sigma$ANs is estimated to be $\pm 50\%$ (one sigma). The largest contribution to the total uncertainty comes from the calculation of $f_{NO}$ for isoprene. Reported uncertainties for the rate constants and radical concentrations involved in $RO_2$ loss (Boyd et al., 2003; Ghosh et al., 2010; Crounse et al., 2011; Peeters et al., 2014) combine to give an overall uncertainty of $\pm 35\%$ in $f_{NO}$ for isoprene. Uncertainty in the values of $\alpha$ ($\pm 25\%$) are also significant contributiors to the total uncertainty. The effects of boundary layer growth are not accounted for, but are unlikely to be important after 10 am (e.g. Xiong et al., 2015). The 50% uncertainty constrains the average $\Sigma$ANs production rate to between 55 and 140 ppt hr$^{-1}$.

Rapid production of $\Sigma$ANs decreases the $NO_x$ lifetime only if it leads to the long-term removal of $NO_x$ from the atmosphere. This can occur either if the alkyl and multifunctional nitrates produced are themselves long lived, or if they have short lifetimes but are lost primarily to deposition or to conversion to a different $NO_z$ species that is long-lived. Despite rapid production of $\Sigma$ANs during the day, the diurnal cycle of $\Sigma$ANs exhibits a decrease between 9 am and 7 pm (Fig. 2c), implying that the $\Sigma$ANs loss rate must be rapid.

While $\Sigma$ANs do not build up over the course of a day, their concentration is strongly correlated with their instantaneous production rate in the afternoon (Fig. 6). We interpret these two results to indicate that $\Sigma$ANs are short-lived and near steady-state in the afternoon. A least-squares fit between $\Sigma$AN production and concentration gives a slope of 1.7 hours and an intercept





of 40 ppt. If ΣANs are near steady-state, then the slope of this correlation is equal to the ΣAN lifetime. The intercept of 40 ppt is interpreted as the large-scale background of long-lived ΣANs during summertime at mid-latitudes. Ethyl and isopropyl nitrate were measured by GC-MS during SOAS, and show a consistent concentration of ∼20 ppt, explaining 50% of the intercept. Previous observations over North America suggest that the summed concentration of other small monofunctional nitrates not measured during SOAS is likely also around 20 ppt in the southeast United States, accounting for the other 50% (e.g. Schneider et al., 1998; Blake et al., 2003; Russo et al., 2010).

A lifetime of 1.7 hours for the reactive component of ΣANs is roughly consistent with previous estimates. Perring et al. (2009) found a lifetime of 1.5–2.5 hr for ΣANs in the southeast United States, based on the correlation between ΣANs and formaldehyde. Multiple studies have also found evidence for rapid loss of ΣANs through particle-phase processing in the southeast United States (e.g. Lee et al., 2015; Pye et al., 2015).

Because most ΣANs are short-lived, they do not serve as a permanent sink of $NO_x$ directly. To establish whether ΣANs are a component of $NO_{SL}$ or $NO_{LL}$, the fate of ΣANs must be understood. Conversion of an alkyl nitrate to another alkyl nitrate does not affect the measurement of ΣANs and therefore does not contribute to the calculated 1.7 hour lifetime. The only other $NO_y$ compounds produced by alkyl nitrate oxidation that have been observed in laboratory experiments are $NO_x$ and $HNO_3$ (e.g. Lee et al., 2014; Darer et al., 2011). These two products are thought to arise from completely different mechanisms in the oxidation of ΣANs. $NO_x$ is produced either during the gas-phase oxidation of nitrates (Lee et al., 2014) or by the photolysis of carbonyl nitrates (Müller et al., 2014), while nitric acid is produced only by the heterogeneous hydrolysis of hydroxy-nitrates (Darer et al., 2011). The question of whether ΣAN to nitric acid conversion is occurring is therefore equivalent to the question of whether deposition and the sum of all gas-phase loss processes are sufficient to explain the 1.7 hr lifetime of ΣANs. If these processes cannot explain the short lifetime of ΣANs, then the unaccounted-for loss is likely due to heterogeneous formation of nitric acid.

An upper limit to the gas-phase oxidation rate of ΣANs can be calculated using measurements of ΣANs by assuming that all alkyl and multifunctional nitrates react with OH and $O_3$ at the same rate as isoprene hydroxy-nitrates and that these reactions all lead to loss of ΣANs. Over three-quarters of the ΣANs produced during SOAS were isoprene hydroxy-nitrates (Fig. 5), making the average loss rate of ΣANs close to the rate for isoprene hydroxy-nitrates. In addition, under low-$NO_x$ conditions, the most likely products of ΣAN oxidation are either $NO_x$ or carbonyl nitrates (Lee et al., 2014). Studies by Müller et al. (2014) indicate that carbonyl nitrates are rapidly photolyzed to release $NO_x$. If the photolysis rate is fast enough, then it is a reasonable approximation to treat ΣANs as releasing $NO_x$ every time they are oxidized.

Only ΣANs present in the gas phase are likely to undergo deposition or reaction with OH or $O_3$. We observe that 30% of ΣANs are in the particle phase during the afternoon; however, even if we assume that all ΣANs are gas-phase, the rate of gas-phase oxidation plus the rate of deposition measured by Nguyen et al. (2015) during SOAS is insufficient to explain the loss of ΣANs in the afternoon (Fig. 7, filled areas).

If the gap between the individual loss processes and the overall loss rate of ΣANs is attributed entirely to ΣAN hydrolysis, then the rate of nitric acid production from ΣANs would be 65 ppt hr$^{-1}$. This is two-thirds of the total ΣAN production rate, and roughly equal to the calculated production rate of tertiary nitrates (Peeters et al., 2014; Rindelaub et al., 2015). Laboratory




experiments have shown that, in general, tertiary nitrates undergo hydrolysis far faster than primary or secondary nitrates (Darer et al., 2011; Hu et al., 2011), making it likely that the rate of $\Sigma$AN hydrolysis is similar to the rate of tertiary $\Sigma$AN production.

While the simultaneous presence of a significant missing source of nitric acid and a missing sink of $\Sigma$ANs supports the idea that $\Sigma$AN to nitric acid conversion is occurring, the missing sink of $\Sigma$ANs is approximately a factor of two larger than the missing source of nitric acid (Fig. 7, hatched area). The discrepancy between the two calculations of the $\Sigma$AN hydrolysis rate could be accounted for by uncertainty in the measurements, in the calculated production rate of $\Sigma$ANs, or in the calculated lifetime of nitric acid. As the data from SOAS is insufficient to determine which of these interpretations is correct, we use the average of the missing $HNO_3$ production rate and the missing $\Sigma$AN loss rate as our best estimate of the $\Sigma$AN hydrolysis rate.

Using this average, the rate of $\Sigma$AN hydrolysis to produce nitric acid is 45 ppt $hr^{-1}$ between 10 am and 2 pm. When this is combined with the loss of $\Sigma$ANs by deposition, 55% of the $\Sigma$ANs produced lead to the permanent removal of $NO_x$ from the atmosphere. Using the hydrolysis rate calculated from only the nitric acid budget or only the $\Sigma$ANs budget changes this fraction to 35 or 75%. The remaining locally produced $\Sigma$ANs are assumed to re-release $NO_x$ back to the atmosphere through oxidation and photolysis.

Based on the lifetime and fate calculated here, locally-produced $\Sigma$ANs have a lifetime to re-release of $NO_x$ of just under 4 hours, making them part of $NO_{SL}$. At the same time, deposition and the rapid conversion of reactive multifunctional nitrates to nitric acid means that the formation of $\Sigma$ANs leads to the significant removal of $NO_{SL}$ from the atmosphere.

### 4.3 Peroxy nitrates

Peroxy nitrates are produced through the association of a peroxy acyl radical with $NO_2$ (R3). While non-acyl peroxy radicals can also associate with $NO_2$, the product is extremely unstable and decomposes within seconds in the summertime boundary layer. Peroxy nitrates are primarily lost by thermal dissociation to form $NO_2$ and a peroxy acyl radical. This acyl radical can either react with $NO_2$ to reform a peroxy nitrate, or react with NO or $HO_2$ to form an acyloxy radical or a peracid. The lifetime of peroxy nitrates therefore depends on the temperature and the relative concentrations of $NO_2$, NO, and $HO_2$ (LaFranchi et al., 2009). Rate constants from Orlando and Tyndall (2012) and Atkinson et al. (2006) for the reactions of peroxy acyl nitrate and acyl peroxy radical were used to calculate the lifetime of peroxy nitrates during SOAS.

During the day, peroxy nitrates re-release $NO_x$ on timescales of 1–2 hours and are a component of $NO_{SL}$. The production of peroxy nitrates therefore does not contribute to the net loss of $NO_{SL}$, but still affects the lifetime of $NO_{SL}$ by adjusting the amount of $NO_x$ available for reactions that produce $\Sigma$ANs or nitric acid. At SOAS, the ratio of peroxy nitrates to $NO_x$ is typically around 0.7 at midday.

There are other loss processes of peroxy nitrates. The reaction of OH with methacryloyl peroxy nitrate (MPAN) is rapid, but MPAN is typically a minor component of total peroxy nitrates (LaFranchi et al., 2009). The deposition rate of peroxy nitrates was not measured during SOAS, but previous measurements in a ponderosa pine forest estimate the deposition velocity to be between 0.5 and 1.3 cm $s^{-1}$ (Wolfe et al., 2009; Min et al., 2012). Using this range of deposition velocities gives a total deposition loss rate of peroxy nitrates of $5 \pm 3$ ppt $hr^{-1}$ in the afternoon.





## 5   The photochemical lifetime of $NO_x$ and $NO_{SL}$

The measured concentrations and calculated production and loss rates of each individual $NO_z$ species can be combined to determine the lifetime of $NO_{SL}$. This lifetime depends on the distribution of $NO_z$ between $NO_{SL}$ and $NO_{LL}$ and the chemical transformations between these two classes. If a 7 hour lifetime to re-release of $NO_x$ is used as the provisional dividing line

between $NO_{SL}$ and $NO_{LL}$, then in the afternoon $NO_{SL}$ was composed of $NO_x$, $\Sigma PNs$, and the reactive component of $\Sigma ANs$. As discussed earlier, both peroxy nitrates and $\Sigma ANs$ have lifetimes to re-release of $NO_x$ of less than 4 hours. During the same time, $NO_{LL}$ was composed of nitric acid and unreactive $\Sigma ANs$. Based on Fig. 6, we assume that there is an average background of 40 ppt of unreactive $\Sigma ANs$, and that all $\Sigma ANs$ greater than this amount are short-lived.

The lifetime of $NO_{SL}$ can then be calculated as $\tau_{NO_{SL}} = [NO_{SL}]/\mathcal{L}(NO_{SL})$. The individual processes that lead to loss of

$NO_{SL}$ and their average value between 10 am and 2 pm during SOAS are shown in Fig. 8. The loss of short-lived reactive nitrogen is dominated by the hydrolysis of $\Sigma ANs$ to produce nitric acid. This single process accounts for 65% of the total $NO_{SL}$ loss.

$NO_{SL}$ is also converted to $NO_{LL}$ during SOAS through the association of OH and $NO_2$ to produce nitric acid and the production of small, unreactive alkyl nitrates. The deposition of both peroxy nitrates and $\Sigma ANs$, as well as the uptake of $NO_x$

by plants, also leads to the loss of $NO_{SL}$. Based on the deposition velocity of $NO_x$ over vegetation measured by Breuninger et al. (2013), the rate of $NO_x$ uptake was calculated to be approximately 1 ppt hr$^{-1}$. A 50% uncertainty in the $\Sigma AN$ hydrolysis rate, combined in quadrature with the uncertainties from the other $NO_{SL}$ loss processes, gives the overall uncertainty in the $NO_{SL}$ loss rate of $\pm$ 25 ppt hr$^{-1}$.

When combined with the average concentration of $NO_{SL}$ of 700 ppt during this same time period, the lifetime of $NO_{SL}$,

and therefore the photochemical lifetime of $NO_x$, is calculated to be $11 \pm 5$ hours. Using any value in this range as the cutoff between $NO_{SL}$ and $NO_{LL}$ does not change the partitioning of $NO_y$ between these two classes.

The long lifetime of $NO_x$ calculated here is qualitatively consistent with the partitioning of $NO_y$ during SOAS. The concentration of $NO_{SL}$ is approximately twice as large as $NO_{LL}$ during the afternoon (Fig. 2). In the absence of large fresh emissions of $NO_x$, this implies that the conversion of $NO_{SL}$ to $NO_{LL}$ must be slow, in agreement with our calculations.

This $NO_x$ lifetime is longer than the lifetime of $NO_x$ calculated in fresh plumes, where observational studies have found lifetimes of 5–8 hours (e.g. Ryerson et al., 1998; Alvarado et al., 2010; Valin et al., 2013). These studies focus solely on the chemistry of $NO_x$ rather than $NO_{SL}$ and recognition of the buffering effect of organic nitrates would extend the lifetimes found in these studies. In addition, the average noontime concentration of OH observed during SOAS was up to a factor of 5 lower than values typically observed in urban areas (e.g. Mao et al., 2010; Rohrer et al., 2014). Lower concentrations of OH slow the

rate of atmospheric oxidation, leading to longer lifetimes of $NO_x$.

If lower OH and the production of $NO_x$ from peroxy nitrates were the only differences between polluted areas and the regional background, then the lifetime of $NO_x$ during SOAS would be significantly longer than 10 hours. However, the production of $\Sigma ANs$ is extremely rapid and the deposition and hydrolysis of these species accounts for the majority of the $NO_x$ removal in this rural region. The VOC mixture present in the southeast United States leads to very high values of OH Reactivity





and $\alpha$, both of which enhance the production of $\Sigma$ANs. High concentrations of VOCs also lead to lower OH concentrations and slower production of nitric acid by reaction (R1). Moving from urban centers to rural or remote regions is therefore also a move from $HNO_3$- to $\Sigma$AN-dominated $NO_x$ chemistry. Changes to our understanding of the production and fate of alkyl and multifunctional nitrates will therefore have a large impact on predictions of the lifetime of $NO_x$ and $NO_{SL}$, with subsequent

impacts on the concentration and distribution of $NO_x$ across a region.

## 6  Conclusions

Measurements in a low-$NO_x$, high-VOC region provide new insights into the lifetime and chemistry of $NO_x$ and $NO_{SL}$ in rural areas. $NO_{SL}$ is found to have an average lifetime of $11 \pm 5$ hours, longer than the lifetimes of $NO_x$ observed in plume studies, which do not account for buffering by short-lived $NO_z$ species. The long lifetime of $NO_{SL}$ makes it relatively evenly

distributed across the region and allows small inputs of $NO_x$ to sustain the concentrations of $NO_{SL}$ observed during SOAS.

The long daytime lifetime of $NO_{SL}$ found here indicates that $NO_x$ emitted on one day will persist into the night where $NO_3$ is often the most important oxidant (Brown and Stutz, 2012). Depending the chemistry taking place, $NO_{SL}$ could either be efficiently removed from the atmosphere at night, or remain in the atmosphere until the next day. Fully understanding the transport and distribution of $NO_x$ across a region therefore requires an understanding of both the daytime and nighttime

chemistry of $NO_x$ and $NO_y$.

The production and loss of $\Sigma$ANs are found to be the most important variables in controlling the lifetime of $NO_{SL}$. $\Sigma$ANs were observed to have a lifetime of under 2 hours during the afternoon. This estimate is in line with many previous estimates of $\Sigma$AN lifetimes, and indicates that $\Sigma$ANs are an important short-lived $NO_x$ reservoir. Observations of both $HNO_3$ and $\Sigma$ANs during SOAS provide strong evidence that both gas-phase oxidation to produce $NO_x$ and particle-phase hydrolysis to produce

nitric acid are important chemical loss processes for $\Sigma$ANs. Comparison of the nitric acid and $\Sigma$AN budgets indicate that between 30 and 70% of the alkyl and multifunctional nitrates produced are converted to nitric acid. Further laboratory and field studies are necessary to better constrain this percentage and to understand the mechanisms that control it.

The vast majority of these $\Sigma$ANs are formed during the oxidation of biogenic hydrocarbons, while much of the $NO_x$ is emitted by anthropogenic activities. In this way, the formation of $\Sigma$ANs represents an important anthropogenic-biogenic

interaction, where the oxidation of biogenic VOCs serves to remove anthropogenic pollution from the atmosphere. In rural and remote regions, the interactions between $NO_y$, $HO_x$, and VOCs are complex and bi-directional. As $NO_x$ emissions decrease, $\Sigma$ANs will likely become an even more important part of the $NO_y$ budget, making it increasingly important that their chemistry and loss be taken into consideration when calculating the lifetime and fate of $NO_x$.

*Acknowledgements.* Financial and logistical support for SOAS was provided by the NSF, the Earth Observing Laboratory at the National
Center for Atmospheric Research (operated by NSF), the personnel at Atmospheric Research and Analysis, and the Electric Power Research Institute. The Berkeley authors acknowledge the support of the NOAA Office of Global Programs (NA13OAR4310067) and the NSF (AGS-

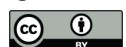



1352972) and by EPA STAR Grant 835407 (to A.H.G.). The Caltech authors acknowledge the support of the NSF (AGS-1331360, AGS-1240604). The Penn State authors acknowledge the support of the NSF (AGS-1246918).



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

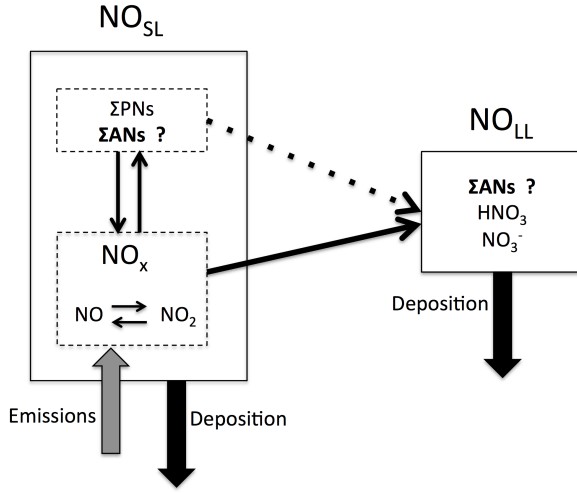

**Figure 1.** A schematic representation of the chemistry of $NO_{SL}$ and $NO_{LL}$, showing the typical components of both classes.

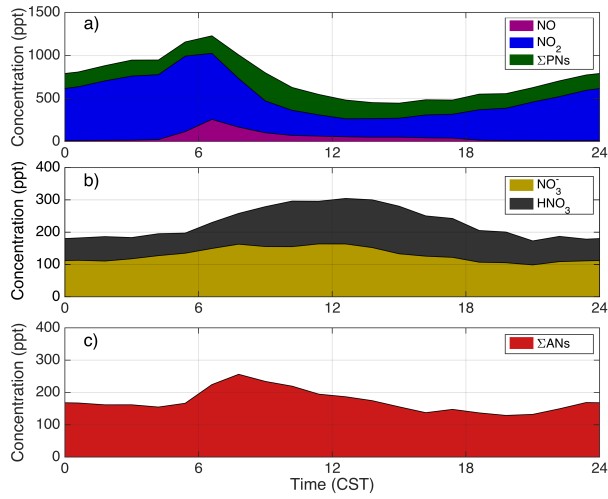

**Figure 2.** Diurnal cycle of measured reactive nitrogen species during SOAS. Reactive nitrogen species are classified as likely components of $NO_{SL}$ (Fig. 2a), likely components of $NO_{LL}$ (Fig. 2b) or unknown (Fig. 2c). The classification into $NO_{SL}$ and $NO_{LL}$ is based on typical summertime afternoon lifetimes.




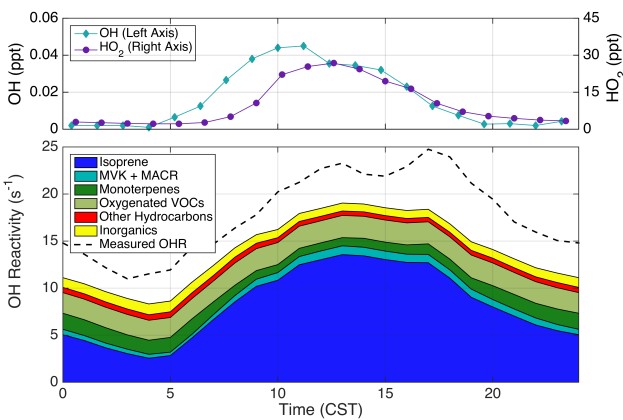

**Figure 3.** Diurnal cycle of $HO_x$ and VOCs during SOAS. The top plot shows the concentration of OH and $HO_2$; the bottom plot shows the VOC Reactivity

.

**Figure 4.** Production rates of $HNO_3$ during SOAS calculated from the reaction of $OH + NO_2$ (black) and inferred from the concentration and deposition rate of $HNO_3$ (blue). The vertical bars show the systematic and random uncertainty in the calculated rates, as described in the text





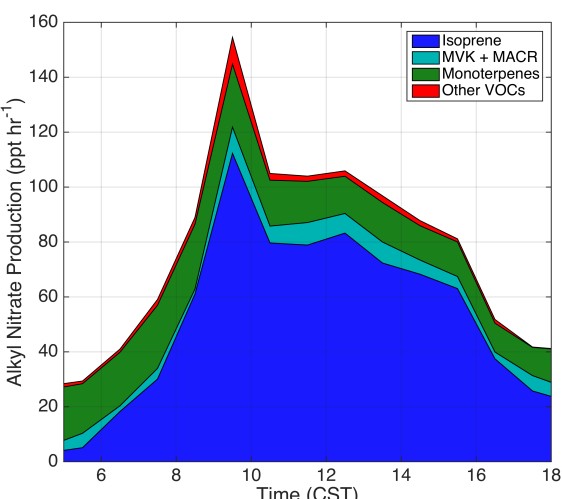

**Figure 5.** Average daytime production of ΣANs, categorized based on VOC precursors

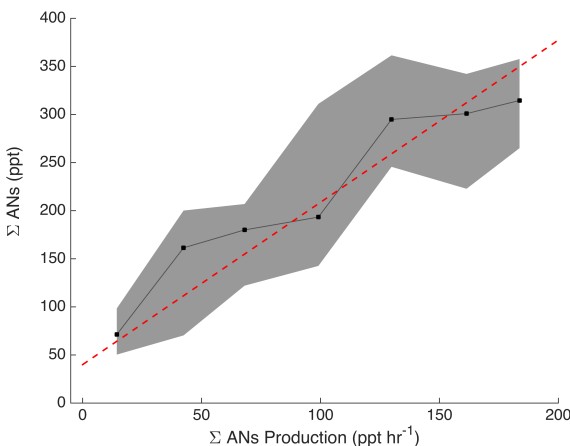

**Figure 6.** The concentration of ΣANs versus their production rate during the afternoon (12 pm – 4 pm). The black squares show the median in each bin, and the shaded grey area the interquartile range. A linear fit gives a slope of 1.7 hr.





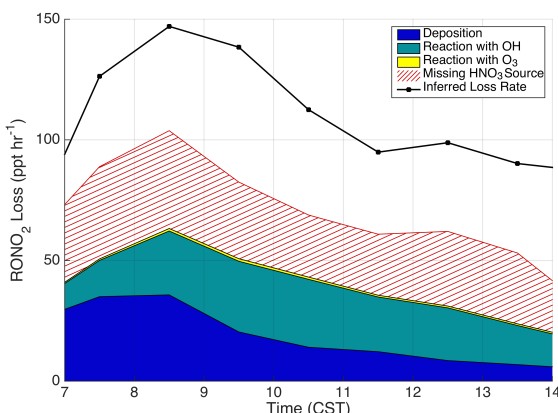

**Figure 7.** Loss rates and fates of $\Sigma$ANs during SOAS. The black line shows the loss rate of $\Sigma$ANs based on the difference between the calculated production rate and the observed change in concentration. The shaded areas show the rates of known $\Sigma$ANs loss processes, and the hatched area shows the missing nitric acid source.

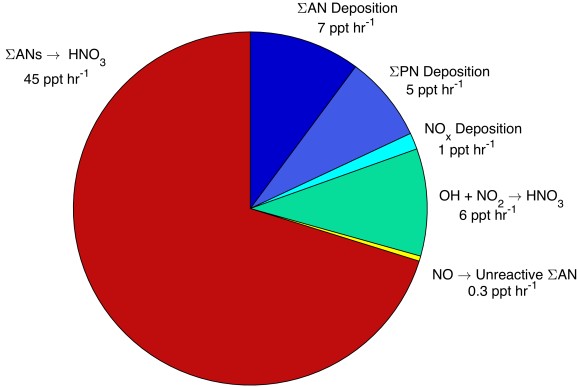

**Figure 8.** The average breakdown of $NO_{SL}$ loss between 10 am and 2 pm during SOAS.