# Peer review of "The Lifetime of Nitrogen Oxides in an Isoprene Dominated Forest"

_Atmospheric Chemistry and Physics, 2016_

## Referee Comment (RC1) · Anonymous Referee #1 · 23 Feb 2016

This paper presents a novel method of assessing the NOx lifetime by using measurements of a range of speciated NOy compounds, taken during a field campaign in rural Alabama, USA. Knowledge of the lifetime of NOx is crucial in understanding concentration and distribution of NOx on a regional and global scale, as well as nitrogen deposition. However, in rural and remote regions, NOx lifetime is often poorly constrained and understood, especially in areas where biogenic VOCs dominate (such as areas of rural USA). The analysis carried out in this paper is very interested and describes a new way of thinking about NOx chemistry that has the potential to be used by other groups in the future. The paper is well written and presented and I would recommend publication in ACP subject to the authors answering the following relatively minor comments.

General comments: P4 line 4: The authors describe how they decide on a cut-off lifetime (of 7 hours) to divide the NOz species between short lived and long lived reactive

nitrogen by using estimated NOx lifetimes determined from satellite observations presented in two cited papers. I realise the details of the estimation are in these papers, however I believe its importance in this manuscript warrants some more discussion here. The authors should explain in more detail how they came to use the 7 hour cut-off.

P5 line 11: Could the authors confirm that this 4th channel on the instrument is the NO3- species plotted in figure 2 – this wasn't clear to me in the manuscript. Again I realise this measurement is reported in other cited literature, however I feel the aerosol phase measurement does warrant some further discussion. For instance, is it purely and aerosol measurement or could there be some interference from HNO3?

P7 line 24: The authors state that the inferred source of HNO3 is in-situ chemical production and not long range transport, due to a lack of large NOx sources near the site and the small variation of the source with wind direction. I am sure this is probably correct however more evidence could be given for this. For instance, in the site description it is stated that the site is 40km SE of Tuscaloosa (population 95000) and 90km SW of Birmingham (population 210000). Do these cities not effect the site at all? Could the authors show a map of the site location with back trajectories to back up their argument?

P8 line 6: Again could transport be considered as a missing source of HNO3 (see comment above).

P12 lines 4 – 8: I am a bit confused by this statement. Surely it is obvious that NO SL will contain the 'reactive' component of ANs and NO LL the unreactive component. What do the authors actually mean by reactive and unreactive components? Could some quantitative measure be put on this and hence what are the type of ANs that make up the two classes?

P13 lines 13 – 15: A statement is made here about both daytime and nighttime chemistry needing to be understood to properly understand the transport and distribution of

NOx, however the analysis concentrates on the daytime chemistry. I feel the conclusions are therefore slightly misleading, unless some nighttime analysis is added to the manuscript.

P13 lines 24 – 25: could a statement be made in the conclusions about how this analysis compares to a more 'classical' analysis of NOx / total NOy (without taking the NOy speciation into account)? This would help cement the importance of the work.

Minor comments / corrections: P3 line 20: It is strange that this is labelled R2b even though it appears first. Should this be re-numbered R2a?

P11 line 16: should 'NO SL' actually be 'NOx'?

At numerous points throughout the manuscript HNO3 and nitric acid are used. The authors should pick one and stick to it throughout.

Figure 3: It maybe of interest to show the O3 average profile as well.

---

## Referee Comment (RC2) · Anonymous Referee #2 · 14 Mar 2016

This is a very well written paper that is worthy of publication in ACP due to its timeliness and high quality. With regard to timeliness, there is a lot of interest in organic-nitrogen in PM and its fate. While I personally believe that calculations such as these are better suited to a 3D model so that many of the assumptions made can be avoided, I doubt that the findings are compromised, and it is likely that results would be within the uncertainty of those presented here. As such, I recommend publication. Only very minor comments follow. In fact, this is likely the fewest number of comments I have ever had to provide on a review!

Minor comments

Chemical formulas (HNO3, RONO2, etc.) are used before they are defined but then defined upon a second use. Either don't define them (assume everyone reading the

paper already knows) or define them upon first use.

Should reaction R3 be shown as reversible as the decomposition can be described using a thermal equilibrium.

Page 4, line 17. Perhaps it would be better to not use primary (as it can connote direct emission), maybe use predominant?

Figure 1 – Please distinguish more strongly between NO3- (nitrate ion) and NO3dot (nitrate radical) as the dash looks like a dot.

Equation 2 – Is the fraction flipped? Should it not be deposition velocity*concentration/BLH?

Page 8, first paragraph... are there any other potential ignored or unidentified HNO3 sources that should at least be mentioned?

Figure 7 – Is photolysis not included because it is assumed to occur so rapidly that it is "included" in the OH reaction (as described in the t

---

## Author Comment (AC1) · 3 May 2016

**Response to Reviewer 1**

**We thank the reviewer for the helpful comments.**

**Reviewer 1**
*P4 line 4: The authors describe how they decide on a cut-off life- time (of 7 hours) to divide the NOz species between short lived and long lived reactive nitrogen by using estimated NOx lifetimes determined from satellite observations pre- sented in two cited papers. I realise the details of the estimation are in these papers, however I believe its importance in this manuscript warrants some more discussion here. The authors should explain in more detail how they came to use the 7 hour cut-off.*

The cut-off between NOsl and NOll is reevaluated using the lifetime of NOx calculated in this paper. Because of this, the choice of 7 hours does not affect the final result. We therefore feel that a detailed analysis of how the 7 hour cut-off was determined is not necessary. We have modified the text to explain how the provisional cutoff was determined and its role in our calculations:

"The division between $NO_{SL}$ and $NO_{LL}$ depends on the lifetime of $NO_x$. For the initial discussion in this manuscript, we use a provisional lifetime of 7 hours to divide $NO_z$ species between $NO_{SL}$ and $NO_{LL}$. This cutoff is in the middle of the range of $NO_x$ lifetimes found in plume studies (e.g. Ryerson et al., 1998; Dillon et al., 2003; Alvarado et al., 2010; Valin et al., 2013). The provisional cutoff chosen as a starting point does not affect the final results."

*P5 line 11: Could the authors confirm that this 4th channel on the instrument is the NO3- species plotted in figure 2 – this wasn't clear to me in the manuscript. Again I realise this measurement is reported in other cited literature, however I feel the aerosol phase measurement does warrant some further discussion. For instance, is it purely and aerosol measurement or could there be some interference from HNO3?*

The $NO_3^-$ species plotted in figure 2 was taken from the measurements made by MARGA, and should be specific to only inorganic nitrate. The TD-LIF measurement of organic nitrate is equally sensitive to nitrates in both the gas and particle phases, but the thermal-dissociation oven was set at a temperature where the measurement was not affected by inorganic nitrate. The TD-LIF measurement of organic nitrates in only the particle phase is not used directly in this manuscript, and we have removed the reference to it. We have revised this portion of Section 3 and the caption of Figure 2 to make these points clear:

"$NO_2$, $\Sigma PNs$, and $\Sigma ANs$ were measured via Thermal Dissociation Laser Induced Fluorescence (TD-LIF), as described by Day et al. (2002). Ambient air was drawn into a multipass White cell, where a 532 nm Nd-YAG laser excited the $NO_2$ molecules, and their fluorescence signal was collected on a photomultiplier tube (Hamamatsu H7421-50). The same instrument was used to measure the sum of peroxy nitrates and the sum of alkyl and multifunctional nitrates by first passing the air through a heated oven, where the organic nitrates dissociated to form $NO_2$. Organic nitrates present in the particle phase undergo evaporation and thermal dissociation in the heated ovens to form $NO_2$. The TD-LIF measurement of $\Sigma ANs$ therefore includes alkyl and multifunctional nitrates in both the gas and particle phases, but does not include $HNO_3$ or particle-phase inorganic nitrate (Day et al., 2002; Rollins et al., 2010)."

"Particle-phase inorganic nitrate ($NO^-_3$) was measured using a Monitor for AeRosols and GAses (MARGA) (Allen et al., 2015). Ambient air was drawn through a rotating wet-walled denuder which collected water soluble gas-phase compounds. Particle-phase compounds were captured by a steam-jet aerosol collector downstream of the denuder. Water soluble ions from both phases were then quantified via ion chromatography. This measurement of $NO^-_3$ is designed to be specific to inorganic nitrate, and is not affected by $\Sigma ANs$ in the particle phase (Allen et al., 2015)."

"Figure 2. Diurnal cycle of measured reactive nitrogen species during SOAS. Reactive nitrogen species are classified as likely components of $NO_{SL}$ (Fig. 2a), likely components of $NO_{LL}$ (Fig. 2b) or unknown (Fig. 2c). The classification into $NO_{SL}$ and $NO_{LL}$ is based on typical summertime afternoon lifetimes. The measurement of $HNO_3$ represents nitric acid in the gas phase, while the measurement of $NO^-_3$ represents inorganic nitrate in the particle phase. The measurement of $\Sigma ANs$ includes alkyl and multifunctional nitrates in both the gas and particle phase."

*P7 line 24: The authors state that the inferred source of HNO3 is in-situ chemical production and not long range transport, due to a lack of large NOx sources near the site and the small variation of the source with wind direction. I am sure this is probably correct however more evidence could be given for this. For instance, in the site description it is stated that the site is 40km SE of Tuscaloosa (population 95000) and 90km SW of Birmingham (population 210000). Do these cities not effect the site at all? Could the authors show a map of the site location with back trajectories to back up their argument?*

*P8 line 6: Again could transport be considered as a missing source of HNO3 (see*

*comment above).*

We find no evidence that these cities impact the inferred production rate. We have added an additional paragraph to section 4.1 to explain our reasoning:

"Since the calculation of the inferred production rate considers only the hour-to-hour change in nitric acid and not its gross concentration, the inferred production rate is not affected by distant nitric acid sources. We find very small (less than 15%) variation in the concentration of $NO_x$ with wind direction and no correlation of the inferred production rate around noon with sulfate (a power plant tracer) or benzene (an urban tracer). As the transport time from these sources to the CTR site is significantly greater than 1 hour, this result is not surprising."

.

*P12 lines 4 – 8: I am a bit confused by this statement. Surely it is obvious that NO SL will contain the 'reactive' component of ANs and NO LL the unreactive component. What do the authors actually mean by reactive and unreactive components? Could some quantitative measure be put on this and hence what are the type of ANs that make up the two classes?*

The relationship between concentration and production of $\Sigma ANs$ (Fig. 6) found here indicates that there is a portion of $\Sigma ANs$ that has a lifetime under 2 hours and a portion that has a significantly longer lifetime, although we cannot determine it precisely. Previous work suggests that small, monofunctional nitrates have atmospheric lifetime of days to weeks, making them likely to be the unreactive component of $\Sigma ANs$. We have added the following text to page 10 to clarify this point:

"This background is likely composed of small monofunctional alkyl nitrates, since these compounds typically have lifetimes of days or weeks in the summertime troposphere (e.g Clemitshaw et al., 1997)."

"This reactive component is likely composed of larger, multi- functional nitrates that can be lost rapidly by oxidation, deposition, or hydrolysis (Lee et al., 2014; Nguyen et al., 2015; Darer et al., 2011)."

We have also revised the statement on page 12 to read:

"We interpret the y-intercept in the correlation between $\Sigma AN$ production and concentration (Fig. 6) to represent a 40 ppt background of unreactive $\Sigma ANs$, likely composed of small monofunctional nitrates. We treat all $\Sigma ANs$ greater than this constant background as short-lived."

*P13 lines 13 – 15: A statement is made here about both daytime and nighttime chemistry needing to be understood to properly understand the transport and distribution of NOx, however the analysis concentrates on the daytime chemistry. I feel the conclu- sions are therefore slightly misleading, unless some nighttime analysis is added to the manuscript.*

Our intention with this statement was explain that, when considering the distribution and transportation of NOx across a region, applying the noontime lifetime of NOx is not sufficient. We have revised the statement as follows:

"To fully understand the transport and distribution of $NO_X$ across a region the daytime chemistry of $NO_X$ discussed here must be combined with additional analyses of the nighttime chemistry of $NO_X$ and $NO_y$ (e.g. Brown et al., 2009; Crowley et al., 2011; Ayres et al., 2015)."

*P13 lines 24 – 25: could a statement be made in the conclusions about how this analysis compares to a more 'classical' analysis of NOx / total NOy (without taking the NOy speciation into account)? This would help cement the importance of the work.*

We do not believe that a calculation of the NOx lifetime using the NOx/NOy ratio is applicable here. Previous analyses doing this have all been in the outflow of a clearly defined plume, which is not the case here. We have added a sentence here explaining this:

"More quantitative calculations of the $NO_x$ lifetime using the ratio of $NO_{SL}$ to $NO_{LL}$ or $NO_x$ to $NO_y$ have been developed for analyses of plumes (e.g. Kleinman et al., 2000; Ryerson et al., 2003) but are not adaptable to this data set."

*Minor comments / corrections: P3 line 20: It is strange that this is labelled R2b even though it appears first. Should this be re-numbered R2a?*

While we appreciate that it is unusual for R2b to appear before R2a, most descriptions of the RO2 + NO reaction (e.g. Perring et al, 2013, Finlayson-Pitts and Pitts, 1999) refer to the channel that forms alkyl nitrates as the 'b' reaction. We have kept this reaction labeled R2b in order to be consistent with these analyses.

*P11 line 16: should 'NO SL' actually be 'NOx'?*

We have revised the sentence to read:

"At the same time, deposition and the rapid conversion of reactive multifunctional nitrates to nitric acid means that the formation of $\Sigma$ANs leads to the significant removal of $NO_{SL}$, and therefore $NO_x$, from the atmosphere."

*At numerous points throughout the manuscript HNO3 and nitric acid are used. The authors should pick one and stick to it throughout.*

We have chosen to use 'nitric acid' throughout the manuscript, except in a few cases where we feel using the chemical formula is more appropriate.

*Figure 3: It maybe of interest to show the O3 average profile as well.*

We have added the diurnal profile of ozone to Figure 3.

---

## Author Comment (AC2) · 3 May 2016

**Response to Reviewer 2**

**We thank the reviewer for the helpful comments.**

**Reviewer 2**

*Chemical formulas (HNO3, RONO2, etc.) are used before they are defined but then defined upon a second use. Either don't define them (assume everyone reading the paper already knows) or define them upon first use.*

We have gone through the manuscript to be consistent in our definitions of chemical formulas. We assume the reader is familiar with simple chemical formulas (e.g. $O_3$, $HNO_3$), but include definitions of classes of compounds that have multiple abbreviations (e.g. $\Sigma ANs/\ \Sigma RONO_2$)

*Should reaction R3 be shown as reversible as the decomposition can be described using a thermal equilibrium.*

We have changed reaction R3 to be reversible

*Page 4, line 17. Perhaps it would be better to not use primary (as it can connote direct emission), maybe use predominant?*

We have changed this sentence to avoid the word primary:

"This is especially true for the multifunctional, biogenically-derived nitrates that are the predominant component of $\Sigma ANs$ in forested areas (e.g. Beaver et al., 2012)."

*Figure 1 – Please distinguish more strongly between NO3- (nitrate ion) and NO3dot (nitrate radical) as the dash looks like a dot.*

We have made the dash in Figure 1 significantly larger to make this clearer.

*Equation 2 – Is the fraction flipped? Should it not be deposition velocity\*concentration/BLH?*

We thank the reviewer for noticing this error. We have corrected this equation to read as:

$$L(HNO3) = \frac{v_{dep}}{BLH} \cdot [HNO3]$$

*Page 8, first paragraph. . . are there any other potential ignored or unidentified HNO3 sources that should at least be mentioned?*

While there are other possible sources of nitric acid, these sources would affect only the

budget of nitric acid and not alkyl nitrates. We have added a statement about these other possible sources to the end of Section 4.1:

"If other processes are responsible for the missing nitric acid source, these would not affect the budget of $\Sigma$ANs. Only the conversion of $\Sigma$ANs to nitric acid will lead to a missing source of nitric acid and a missing sink of $\Sigma$ANs."

*Figure 7 – Is photolysis not included because it is assumed to occur so rapidly that it is "included" in the OH reaction*

This is correct. We have revised the text to clarify this point:

"Since isoprene hydroxy-nitrates and most other first-generation nitrates must be further oxidized before undergoing photolysis, we do not include photolysis as a separate loss process in Fig. 7. Nitrates produced in the oxidation of compounds such as MVK and MACR can undergo photolysis without reacting with OH first, but these are a minor fraction of the total $\Sigma$AN production rate (Fig. 5)."